

# Immune priming against bacteria in spiders and scorpions?

Dumas Gálvez[1,2], Yostin Añino[3], Carlos Vega[4] and Eleodoro Bonilla[4]

[1] Programa Centroamericano de Maestría en Entomología, Universidad de Panamá, Panama, Panama
[2] COIBA AIP, Panama, Panama
[3] Museo de Invertebrados G.B. Fairchild, Universidad de Panamá, Panama, Panama
[4] Escuela de Biología, Universidad de Panamá, Panama, Panama

Corresponding author
Dumas Gálvez,
dumas.galvezs@up.ac.pa

## ABSTRACT

Empirical evidence of immune priming in arthropods keeps growing, both at the within- and trans-generational level. The evidence comes mostly from work on insects and it remains unclear for some other arthropods whether exposure to a non-lethal dose of a pathogen provides protection during a second exposure with a lethal dose. A poorly investigated group are arachnids, with regard to the benefits of immune priming measured as improved survival. Here, we investigated immune priming in two arachnids: the wolf spider *Lycosa cerrofloresiana* and the scorpion *Centruroides granosus*. We injected a third of the individuals with lipopolysaccharides of *Escherichia coli* (LPS, an immune elicitor), another third were injected with the control solution (PBS) and the other third were kept naive. Four days after the first inoculations, we challenged half of the individuals of each group with an injection of a high dose of *E. coli* and the other half was treated with the control solution. For scorpions, individuals that were initially injected with PBS or LPS did not differ in their survival rates against the bacterial challenge. Individuals injected with LPS showed higher survival than that of naive individuals as evidence of immune priming. Individuals injected with PBS tended to show higher survival rates than naive individuals, but the difference was not significant—perhaps suggesting a general immune upregulation caused by the wounding done by the needle. For spiders, we did not observe evidence of priming, the bacterial challenge reduced the survival of naive, PBS and LPS individuals at similar rates. Moreover; for scorpions, we performed antibacterial assays of hemolymph samples from the three priming treatments (LPS, PBS and naive) and found that the three treatments reduced bacterial growth but without differences among treatments. As non-model organisms, with some unique differences in their immunological mechanisms as compared to the most studied arthropods (insects), arachnids provide an unexplored field to elucidate the evolution of immune systems.

## INTRODUCTION

The invertebrate immune system was traditionally believed to contain no memory and specificity. This is due to the lack of immune machinery that is needed in order to develop the desired immune response in vertebrates (*Rowley & Powell, 2007*). However, recent literature has reported that invertebrates exposed to a low dose of a pathogen can

obtain protection against a subsequent lethal dose of the same pathogen, a phenomenon termed as immune priming (*Little & Kraaijeveld, 2004*). This improved immune response can be observed within a few days after the priming, in later stages of the individual ('within-generation immune priming', *Milutinović & Kurtz, 2016*) or even transferred to the offspring ('trans-generational immune priming', *Tetreau et al., 2019*).

Evidence of immune priming comes mostly from insects (*Milutinović et al., 2016*; *Cooper & Eleftherianos, 2017*). However, a number of studies found no evidence of immune priming in insects against fungi (*Reber & Chapuisat, 2012*; *Gálvez & Chapuisat, 2014*), bacteria (*González-Tokman et al., 2010*; *Patrnogic et al., 2018*) and bacterial immune elicitors (*ter Braak et al., 2013*; *Wu et al., 2015b*). In some cases the detection depended on the pathogen used (*Pham et al., 2007*; *Vargas et al., 2016*; *Ferro et al., 2019*). Overall, the detection of immune priming in insects seems to depend on multiple factors such as host-pathogen combination, host lifespan, priming method, pathogen dose, virulence, among others (*Contreras-Garduño, 2016*; *Milutinović et al., 2016*; *Cooper & Eleftherianos, 2017*; *Tetreau et al., 2019*).

Even though it is thought that the immune system of arthropods is well conserved across species, based on an innate immune system, consisting of cellular and humoral responses (*Rowley & Powell, 2007*), recent studies showed there exists some variation across taxa and the insect immune system that which does not necessarily characterize other arthropods. For instance, *Bechsgaard et al. (2016)* discovered that some genes involved in pathways for pathogen recognition (e.g., bacteria) have been lost in arachnids and the humoral immune effector proteins (antimicrobial peptides, AMPs) are apparently not induced as it is the case for insects, but they are constitutively produced, a trend also observed by previous studies (*Lorenzini et al., 2003*; *Fukuzawa et al., 2008*; *Baumann et al., 2010*; *González-Tokman et al., 2014*). In other arachnids, the evidence seems to suggest a complete absence of an induced immune response (*Santos-Matos et al., 2017*). Another example of dissimilarities between insects and arachnids is the evidence indicating that phagocytosis plays a role in the immune priming of insects (*Pham et al., 2007*; *Wu et al., 2015a*). However, in spiders, phagocytosis seems to play a minor role in defense when compared to AMPs and coagulation (*Fukuzawa et al., 2008*). Overall, whether these differences in arachnids' immune systems influence their capacity to mount an immune priming response is unclear.

Immunological studies and evidence of immune priming in arachnids come mainly from work with ticks, given their medical importance, with evidence of upregulation (*Nakajima et al., 2001*; *Matsuo et al., 2004*) and improved survival after exposure to an immune elicitor, controlled by molecular pathways that are apparently unique to ticks (*Shaw et al., 2017*). Moreover, blood-feeding can strongly upregulate defensin genes in the midgut, which normally occurs in the fat body after bacterial infection in insects (review in *Taylor, 2006*). Ticks as hematophagous are an atypical group of arachnids in terms of the use of immune defenses; for instance, ticks can use fragments of the host blood for their own defense against bacteria in the midgut level (*Nakajima et al., 2003*; *Nakajima et al., 2005*), together with their own antibacterial peptides (*Nakajima et al., 2005*) or with the influence of commensal and symbiont bacteria (*Chávez et al., 2017*). In contrast, knowledge about the immune system of other arachnids remains mostly unknown.

In fact, no experimental study has investigated immune priming in terms of increased survival in non-hematophagous arachnids like spiders or scorpions (*Milutinović & Kurtz, 2016*; *Milutinović et al., 2016*). By studying the immune response of other arachnids, analogies and differences with other taxa can be established in order to understand the evolution of the immune systems in invertebrates. Here, we performed the first test of immune priming in spiders and scorpions in terms of improved survival. We investigated whether the wolf spider *Lycosa cerrofloresiana* (Lycosidae) and the scorpion *Centruroides granosus* (Buthidae) can mount an immune priming response when injected with lipopolysaccharides (LPS) of *Escherichia coli* and subsequently challenged with a lethal dose of the same bacteria. If antimicrobial peptides are constitutively produced, then their immune system may always be prepared for an immune challenge and exposure to a priming agent may not be required. Alternatively, priming would both trigger the release of constitutive components and induce recruitment of production of higher levels of antimicrobials components.

## MATERIALS & METHODS

### Study species

This study was carried out with two nocturnal terrestrial predators, the wolf spider *Lycosa cerrofloresiana* Petrunkevitch, 1925 and the scorpion *Centruroides granosus* Thorell, 1876 (Buthidae). *Lycosa cerrofloresiana* is found from El Salvador to Panama (*World Spider Catalog, 2019*), while *C. granosus* is endemic to Panama (*De Armas, Teruel & Kovařík, 2011*). For both species, all the existing literature is on aspects of taxonomy and distribution (e.g., *De Armas, Teruel & Kovařík, 2011*; *World Spider Catalog, 2019* and references therein). Still, *Centruroides granosus* prey on a variety of arthropods, including insects and other arachnids (*Miranda et al., 2015*). Literature on the diet of the wolf spider is missing but we have noticed spiders eating crickets and cockroaches in the field.

Spiders were collected from a baseball field in the town of Gamboa (09°07′05.1596″, −079°42′03.5266″) and scorpions were collected from a dirt road in the town of Polanco (08°45′44.3196″, −079°48′22.8618″). All individuals were fed with the cricket *Acheta domesticus*, one week before the experiments. The study did not involve unethical handling of animals and did not require permits for experimentation by the Bioethics Office from the University of Panama. We collected all specimens under the collection permit SE/AH-2-18 issued by the 'Ministerio de Ambiente', the government entity in charge of the management of natural resources.

### Immune priming

A strain of *Escherichia coli* was used for the experiments, which was obtained through isolation with selective media by the Department of Microbiology of the Biology School at the University of Panama. Tests of virulence of this strain produced high mortality in both spiders and scorpions (Supplementary material). Previous studies have used *E. coli* via injection or pricking as an immune elicitor in other arthropods (*Eleftherianos et al., 2006*; *Roth & Kurtz, 2009*; *Erler, Popp & Lattorff, 2011*; *Santos-Matos et al., 2017*) and arachnids (*Sonenshine et al., 2003*; *Santos-Matos et al., 2017*).

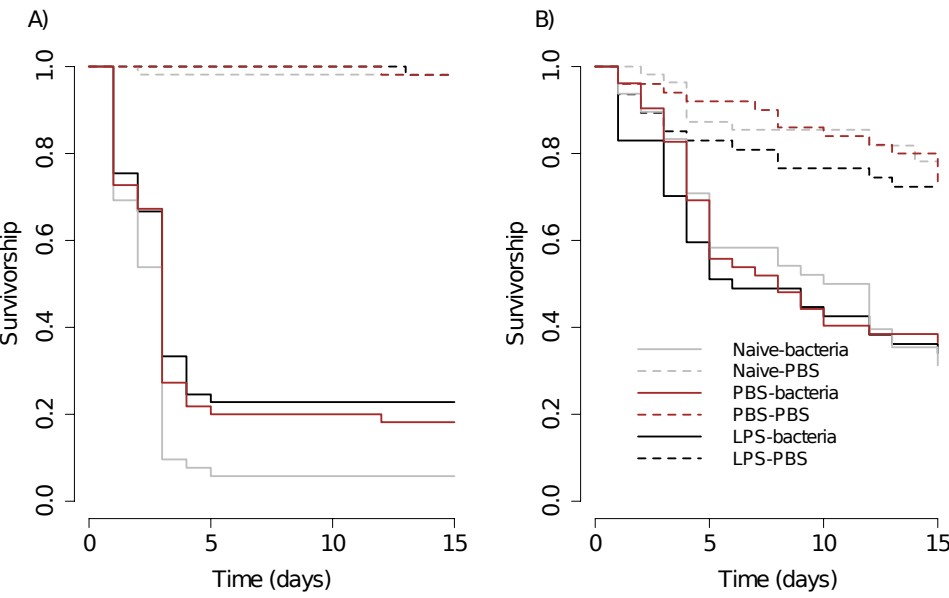

**Figure 1  Kaplan–Meier survival curves of scorpions (A) and spiders (B), under different priming treatments.** After the priming period, half of the individuals of each treatment were injected with the control solution (Naive –PBS, PBS –PBS and LPS –PBS) or with the bacterial solution (Naive –Bacteria, PBS –Bacteria, LPS –Bacteria). Scorpions: Naive –PBS, $n = 54$; Naive –Bacteria, $n = 52$; PBS –PBS, $n = 53$; PBS –Bacteria, $n = 55$; LPS –PBS, $n = 51$; LPS –Bacteria, $n = 57$. Spiders: Naive –PBS, $n = 55$; Naive –Bacteria, n =48; PBS –PBS, $n = 50$; PBS –Bacteria, $n = 52$; LPS –PBS, $n = 47$; LPS –Bacteria, $n = 47$. See text for statistical details.

We used chilling anesthesia for all injections, which consisted of placing scorpions and spiders at 4 °C for 20 min. In order to stimulate priming, we injected spiders with 138 nL of LPS in PBS (0.5 mg / mL; Sigma: L8274; hereafter LPS) by using a Nanoliter 2010 injector (WPI, Florida, USA). For scorpions, we picked 100 µL of the LPS solution with a micropipette to fill insulin syringes used for the injections. Control groups consisted of individuals injected only with PBS and another group of untreated individuals (naive) to test whether the mechanical damage caused by the injections was enough to prime the immune system. For spiders, the injection procedure during the priming caused around 1% mortality and there was no mortality in scorpions.

For the bacterial challenge, bacteria were cultured overnight on lysogeny broth (LB) at 27 °C. We centrifuged 14 ml of the culture (LD$_{50}$ $1 \times 10^7$ cells / mL) at 4,000 rpm for 5 min, the pellet was washed with PBS and resuspended in 14 ml of PBS. Four days after the initial injections, half of the individuals in each treatment were injected with the bacterial solution (138 nL for spiders and 100 µL for scorpions; Naive –Challenged, PBS –Challenged, LPS –Challenged, see Fig. 1 for details on sample sizes). As controls, the other half of the individuals of each treatment were injected only with PBS (138 nL for spiders and 100 µL for scorpions; Naive –Control, PBS –Control, LPS –Control, see Fig. 1 for details on sample sizes). We performed the experiments twice, on separate dates and monitored the survival of spiders and scorpions for 15 days after the final challenge.

## Antibacterial activity

For these measurements, we were only able to collect sufficient hemolymph samples from individual scorpions. To test whether the priming with LPS upregulated the production of antimicrobial components found in the hemolymph, we measured antibacterial activity following a protocol modified from *Wu et al. (2014)*. Three days after the priming phase, we collected 10 µL of hemolymph from each treatment (Naive: $n = 9$; PBS: $n = 6$ and LPS: $n = 9$) by pricking chilled animals and placed it immediately in ice and later stored at −20 °C. The antibacterial test consisted of mixing 10 µL of cell-free hemolymph (centrifuged at 4,000 rpm for 5 min) with 10 µL of *E. coli* culture ($1 \times 10^7$ cells / mL) in 180 µL of LB and incubated during 14 h at 27 °C in 1.5 mL Eppendorf tubes.

Antibacterial activity was quantified as inhibition of bacterial growth in the samples by measuring optical density at 630 nm on a 96-well microplate reader. To evaluate whether the hemolymph samples inhibited the bacterial growth, we used a positive control in which we placed 10 µL of *E. coli* culture in 190 µL of LB (three replicates).

## Statistical analysis

All analyses were performed in R (*R Development Core Team, 2019*). The Kaplan–Meier survival analysis was carried out to test for differences in survival rates between treatments as implemented in the package 'survival'. Moreover, we tested for differences between sexes in both species as a fixed factor. We used the Gehan-Breslow-Wilcoxon test to compare survival rates across treatments at early time points and the log-rank test to compare treatments at the end of the experiments (package survMisc). For the antibacterial activity, we performed a one-sample Wilcoxon test for each treatment to assess whether the priming treatment reduced bacterial growth as compared to the mean bacterial growth in the absence of hemolymph ($OD_{630} = 0.763$). To compare treatments, we carried out a Kruskal-Wallis test.

# RESULTS

## Immune priming

For scorpions, overall, sex has no effect on survival (log-rank: $z = -0.04$, $p = 0.97$). The bacterial challenge significantly reduced the survival in each treatment (Naive - Bacteria vs Naive –PBS, PBS - Bacteria vs PBS –PBS, LPS - Bacteria vs LPS –PBS, Fig. 1A, Table 1). We found evidence of immune priming because scorpions initially injected with LPS showed higher levels of survival against the bacterial challenge than that of naive scorpions (LPS - Bacteria vs Naive - Bacteria, Fig. 1A, Table 1). Although the results suggests that the priming could be elicited by the wounding caused by the injection, this trend was not significant overall (PBS –Bacteria vs Naive –Bacteria, Table 1) and neither during the early stages of the infection (Gehan-Breslow-Wilcoxon test in Table 1).

The survival between scorpions injected initially with PBS or LPS against the bacterial challenge was not significantly different (PBS –Bacteria vs LPS - Bacteria, Fig. 1A, Table 1). The survival of controls of the three treatments were not significantly different (Naive - PBS vs PBS –PBS, Naive –PBS vs LPS –PBS, PBS –PBS vs LPS - PBS, Fig. 1A, Table 1).

**Table 1 Survival analysis pairwise comparisons of priming treatments exposed to a control solution (-PBS) or to a bacterial challenge (- Bacteria).** The Gehan-Breslow-Wilcoxon test compares survival rates at early time points and the Logrank tests compares them at late time points. See Materials and Methods for details on priming treatments.

| Comparison | Gehan-Breslow-Wilcoxon | Log-rank |
|---|---|---|
| *Scorpions* | | |
| Naive - Bacteria vs Naive - PBS | $z = -7.8, p < 0.001$ | $z = 4.1, p < 0.001$ |
| PBS - Bacteria vs PBS - PBS | $z = -8.3, p < 0.001$ | $z = 4.3, p < 0.001$ |
| LPS - Bacteria vs LPS - PBS | $z = -7.8, p < 0.001$ | $z = 4.1, p < 0.001$ |
| LPS - Bacteria vs Naive - Bacteria | $z = 1.9, p = 0.05$ | $z = -3.65, p = 0.03$ |
| PBS - Bacteria vs Naive - Bacteria | $z = 1.9, p = 0.11$ | $z = -3.65, p = 0.07$ |
| PBS - Bacteria vs LPS - Bacteria | $z = 0.38, p = 0.69$ | $z = -0.63, p = 0.53$ |
| Naive - PBS vs PBS - PBS | $z = 0, p > 0.05$ | $z = 0.01, p = 0.99$ |
| Naive - PBS vs LPS - PBS | $z = -0.02, p = 0.98$ | $z = -0.03, p = 0.98$ |
| PBS - PBS vs LPS - PBS | $z = -0.01, p = 0.98$ | $z = -0.03, p = 0.98$ |
| *Spiders* | | |
| Naive - Bacteria vs Naive - PBS | $z = -4.7, p < 0.0001$ | $z = -4.06, p < 0.0001$ |
| PBS - Bacteria vs PBS - PBS | $z = -4.0, p < 0.0001$ | $z = -3.7, p < 0.001$ |
| LPS - Bacteria vs LPS - PBS | $z = -3.4, p < 0.001$ | $z = 3.4, p < 0.001$ |
| LPS - Bacteria vs Naive - Bacteria | $z = -0.75, p = 0.44$ | $z = 0.27, p = 0.79$ |
| PBS - Bacteria vs Naive - Bacteria | $z = -0.06, p = 0.95$ | $z = -0.12, p = 0.9$ |
| PBS - Bacteria vs LPS - Bacteria | $z = -0.80, p = 0.42$ | $z = 0.41, p = 0.68$ |
| Naive - PBS vs PBS - PBS | $z = 0.53, p = 0.59$ | $z = 0.61, p = 0.54$ |
| Naive - PBS vs LPS - PBS | $z = -0.91, p = 0.36$ | $z = -0.78, p = 0.44$ |
| PBS - PBS vs LPS - PBS | $z = -0.34, p = 0.73$ | $z = -0.19, p = 0.85$ |

For spiders, the influence of sex on survival was investigated in the first trial and was not significant ($z = -1.89, p = 0.06$). The bacterial challenge significantly reduced the survival of all the priming treatments (Naive - Bacteria vs Naive –PBS, PBS - Bacteria vs PBS –PBS, LPS - Bacteria vs LPS –PBS, Fig. 1B, Table 1). The three priming treatments did not vary in the survival against the bacterial challenge (Naive - Bacteria vs PBS –Bacteria, Naive –Bacteria vs LPS –Bacteria, PBS –Bacteria vs LPS - Bacteria, Fig. 1B, Table 1). The controls of the three priming treatments were not significantly different (Naive - PBS vs PBS –PBS, Naive –PBS vs LPS –PBS, PBS –PBS vs LPS - PBS, Fig. 1B, Table 1).

## Antibacterial activity

Hemolymph of naive scorpions inhibited *E. coli* growth when compared to the average growth of the bacteria without hemolymph (Wilcoxon: $V = 0, p = 0.002, n = 9$, Fig. 2) as well as the hemolymph of scorpions injected with PBS (Wilcoxon: $V = 2, p = 0.05, n = 6$, Fig. 2) and the hemolymph of scorpions injected with LPS (Wilcoxon: $V = 6, p = 0.03, n = 9$, Fig. 2). Overall, there were no differences between priming treatments in their capacity to inhibit bacterial growth (Kruskal–Wallis: $X^2 = 0.27, d.f. = 2, p = 0.87$, Fig. 2).

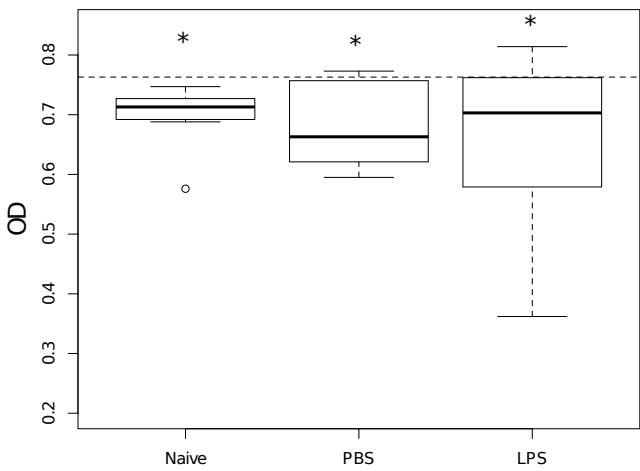

**Figure 2** **Growth in vitro of *Escherichia coli* when mixed with hemolymph samples of scorpions from the different priming treatments, measured as optical density (OD).** Stars indicate treatments that significantly reduced the bacterial growth as compared to the bacterial growth in the absence of any hemolymph (dashed line, $OD_{630} = 0.763$). Overall, treatments did not differ in their capacity to inhibit the bacterial growth. Naive: $n = 9$; PBS: $n = 6$ and LPS: $n = 9$.

## DISCUSSION

Scorpions as organisms with relatively long lifespans (*Lourenço, 2000*) are more likely to be exposed to a pathogen multiple times during their lifetime; therefore, they are good candidates to show immune priming (*Best et al., 2013*). Indeed, we found evidence of immune priming in terms of improved survival for individuals that were treated with LPS as compared to naive individuals. It is unclear whether wounding by itself is sufficient to elicit priming since control individuals (injected with PBS) showed similar survival against the bacteria to individuals injected with LPS or kept naive. Thus, further work should evaluate whether wounding may be sufficient to trigger priming in arachnids as seen in other arthropods (*Korner & Schmid-Hempel, 2004*; *Roth et al., 2010*; *Nam et al., 2012*). Perhaps danger-associated molecular patterns (DAMPs) associated to wound healing could trigger immune priming (*Krautz, Arefin & Theopold, 2014*) or they may allow the entrance of pathogens that trigger the priming.

The presence of LPS in the hemolymph should have triggered the production of AMPs (*Rodríguez De La Vega et al., 2004*) or other antimicrobial effectors; however, our antibacterial activity assay with scorpions' hemolymph suggests that there was no upregulation of AMPs in primed individuals, in line with previous work in scorpions comparing control and challenged individuals (*Cocianich et al., 1993*; *Ehret-Sabatier et al., 1996*). However, the freezing and thawing of the samples may have influenced the antibacterial effect, as it was not a part of the original protocol or perhaps the detection of an effect requires larger sample sizes. Another concern is that the immunological history of the individuals used for experimentation was unknown (e.g., priming occurring before the experiments) and whether this influences the immune priming response. Future studies

should try to establish potential model species that could be reared in the laboratory for immunological studies.

The improved resistance by priming may result from other factors or in interaction with AMPs in the hemolymph, which might not perform well in the medium used for our assay. *Rodríguez De La Vega et al. (2004)* found in *Centruroides limpidus* the existence of inducible AMPs and proposed a cooperative antibacterial activity with constitutive hemolymph components. Still, the differences between the survival experiment and the antibacterial activity illustrate how disease resistance and immunity assays may not correlate or are pathogen dependent (review in *Adamo, 2004*); consequently, providing different resolutions to the experimental detection of immune priming in arthropods. Furthermore, assays developed for insects may not be appropriate for arachnids as pointed out by other studies (*Gilbert, Karp & Uetz, 2016*). Future studies should investigate the efficacy of different methods to measure immune components in arachnids.

In spiders, *Gilbert, Karp & Uetz (2016)* provided some indirect evidence of immune priming, finding in the wolf spider *Schizocosa ocreata* that juveniles fed with another gram-negative pathogenic bacteria showed higher encapsulation response against a nylon monofilament implant in the adult stage. In contrast, we did not find benefits in terms of increased survival for wolf spiders that were 'primed' and challenged in the adult stage, suggesting that the age in which priming occurs should be examined. Future studies on arachnids should be aimed at identifying mechanisms, including multiple host –pathogen or host - elicitor (e.g., dead pathogen, other molecules) combinations to evaluate specificity, duration, the effect of symbionts or other potential influential factors. For example, the mode of infection: *Keiser et al. (2016)* showed that a bacterial cocktail increased mortality of a social spider via cuticular topical application while on the contrary spiders fed with crickets injected with the same bacterial cocktail showed longer lifespans than spiders fed with control crickets.

Arachnids offer systems to study other means of defense against pathogens. For instance, the silk of spiders can have antibacterial properties (*Wright & Goodacre, 2012*) and cuticular antifungals have been found in subsocial spiders (*González-Tokman et al., 2014*). In addition, there is extensive evidence revealing AMPs in the venom of spiders and scorpions that are active against bacteria, fungi, viruses and parasites in vitro, which is being aimed at medical applications (*Santos, Reis & Pimenta, 2016*; *Wang & Wang, 2016*). However, we are not aware of studies that investigated the venom - immune system interaction in arachnids when coping with pathogens. Our priming procedure and lethal injection did not allow the interaction between the venom and the bacteria. One might expect that the deactivation of the bacteria by the venom inoculated in the prey may generate a form of priming agent (e.g., dead bacteria) that would act after ingestion.

Despite the inherent differences in the immune system of insects and spiders, immune priming seems to be conserved as a general protection mechanism across arthropods taxa. As non-model organisms, arachnids provide alternative systems to study the evolution of immune systems in non-vertebrate animals and our study adds support to the hypothesis that all organisms should have some sort of acquired immunity (*Rimer, Cohen & Friedman, 2014*).

## CONCLUSIONS

The aim of the study was to test whether immune priming occurred in two arachnid species: a scorpion and a wolf spider. Injection of bacterial components (LPS) seemed to trigger the immune system of the scorpions as they showed improved survival against alive bacteria as compared to individuals that remained untreated (naive). However, scorpions injected with LPS showed similar survival rates as scorpions injected with only a saline solution (PBS), suggesting that the damage caused by injection may be enough to trigger the upregulation of the immune system. The lack of differences in antibacterial assays with scorpions' hemolymph from the different treatments; together with the lack of evidence for immune priming in spiders, it indicates that the experimental detection of this phenomenon may depend on multiple variables (host –pathogen, priming method, host lifespan, virulence, among other) as proposed in the literature.

## ACKNOWLEDGEMENTS

We thank Griselda Arteaga from the Faculty of Medicine at the University of Panama for facilitating the use of the microplate reader. We also thank Alexandre Chausson for his comments on the manuscript; David Camacho, Randhy Rodríguez and Bastien König for their help in the field and laboratory.

### Funding

This study was supported by the SENACYT with the grant APY-NI-2017b-13 for Yostin Añino. The funders had no role in study design, data collection and analysis, decision to publish, or preparation of the manuscript.

### Grant Disclosures

The following grant information was disclosed by the authors:
SENACYT: APY-NI-2017b-13.

### Competing Interests

The authors declare there are no competing interests.

### Author Contributions

- Dumas Gálvez conceived and designed the experiments, performed the experiments, analyzed the data, prepared figures and/or tables, authored or reviewed drafts of the paper, and approved the final draft.
- Yostin Añino, Carlos Vega and Eleodoro Bonilla conceived and designed the experiments, performed the experiments, authored or reviewed drafts of the paper, and approved the final draft.

### Field Study Permissions

The following information was supplied relating to field study approvals (i.e., approving body and any reference numbers):

Specimens collections were authorized by the 'Ministerio de Ambiente' under the permit SE/AH-2-18.

## Data Availability

Raw data are available in the Supplemental File.

## Supplemental Information

Supplemental information for this article can be found online at http://dx.doi.org/10.7717/peerj.9285#supplemental-information.

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
