# Peer review of "Immune priming against bacteria in spiders and scorpions?"

_PeerJ, doi:10.7717/peerj.9285_

## Round 0.1 · original submission · Minor Revisions

Thank you for your study and thank you for your patience in awaiting a review during these troubling times.

The reviewers both agree that this paper needs some revisions. Please address the comments from the two reviewers where possible and add to the discussion any points that need further clarification.

Thank you.

Reviewer 1 ·

Basic reporting

Largely clear English although some areas where it could be improved to aid clarity. Suitable referencing and background.

Experimental design

Design is sound and analyses seem appropriate. Methods could do with more clearly pointing out sample size.

Validity of the findings

Useful results that help us understand how general the process of immune priming is.

Additional comments

Galvez et al provide an interesting study exploring whether there is evidence for immune priming in two different arachnids. This field is very excitign but also suffers from a large body of unreported negative results or challenging results, so I personally welcome stories that illustrate that it is complicated and may not always be there. By at large, I think the study is well conducted and should contribute to the field. I would say, that it can be tough to know why priming isn't seen, or why it isn't seen beyond the priming to wounding. I wonder, for instance, whether LPS is sufficent to induce priming in this system? Would heat inactivated bacterial challenge produce a stronger effect? But that is speculation on my part.

There are some places where writing could be tightened. eg. Sentences beginning intro (L38, L43. L59) which are a bit awkward.

I'm generally not a huge fan of titles as questions. But, that's stylistic, and the story isn't quite clear enough to definitively say yes or no.

Would like to have seen sample size more clearly in methods although it's in the figure legend and is good number of samples per treatment.

The effect of bacterial growth in media with hemolymph is a bit tougher to interpret. There are very small samples, and there might be a slight tendency for stronger inhibition in LPS and PBS injected scorpions but that's just eyeballing the data.

Generally pleased to have more data for different group of arthropods. The authors rightly point out that much of the work on priming is in insects. There are however papers on isopods, crustaceans, nematodes (which are mentioned in intro) but abstract does not suggest this (L15).

L62: wouldn't suggest that constitutive vs induced are an either or situation.

L77: (reviewed in FULL REF)
L79: bacteria at the midgut level -> bacteria in the midgut
L88: in terms of improved survival, we -> improved survival. We

L102-103: is nocturnality relevant?
L128 and around, would be good to see sample sizes here (or at least refer to fig for them?)
L137: N here too

I'm rarely in favor of tables, but here it may work. There are many contrasts with different test statistics etc. These might work a bit better as a table as it can get hard to follow.

·

Basic reporting

The language throughout is precise and easy to follow. In some cases, prepositions are not correctly used (e.g. l51 and l53 should be “dependend on”) and there are a few typos (e.g. “ellicitor” in the abstract). The introduction gives a good background and cites the relevant literature. Maybe, some more information on the life history and lifestyle of the used arachnids could be given. Because they are non-model organisms many readers would probably need some more information. The overall structure is very clear, but the statistical results would be better presented in a table as they impede the reading. The figures are well presented. Figure 1 is missing labels at the two panels (A and B). The raw data seem to be missing. I was only able to find data for the dose response curve in the supplements.

Experimental design

The authors tested two non-model arachnids for within-generation immune priming for the first time and observed acquired immunity in scorpions but not in spiders. The experiments were thoroughly and well conducted and provide us with new insights into an underrepresented part of the field. Methods are described sufficiently.

Validity of the findings

The statistical analysis seems sound, but raw data are missing to confirm. The interpretation of the results makes sense and links back to the research question.

Additional comments

This is one of the first studies into a new field of research, i.e. priming in arachnids and therefore it is understandable that it is somewhat limited. However, I feel that the addition of a second pathogen might enable the authors to generalise their results. Furthermore, as I understand, the animals used here were individuals collected from the wild. The authors therefore do not know the animals' immunological history, i.e. the possibility of priming already happening before the experiment. Eventhough, it might not be possible to repeat experiments or to rear populations in the lab (especially given the currently ongoing pandemic), these issues should at least be mentioned in the discussion.

---

## Round 0.2 · accepted · Accept

Thank you for addressing the reviewer's concerns.

·

Basic reporting

-

Experimental design

-

Validity of the findings

-

Additional comments

The authors have adressed all points. Everything looks good. I don't have anything to add.